# Neurological Manifestations and Clinical Outcomes of Patients with COVID-19 in the Aseer Region, Saudi Arabia

**DOI:** 10.3390/ijerph20053848

**Published:** 2023-02-21

**Authors:** Nada N. Alshehri, Maha A. AlQahtani, Fatima Riaz, Syed E. Mahmood, Ausaf Ahmad, Nawal F. AbdelGhaffar, Abdulaziz H. Abdullah Almakfor, Fawziah M. Alahmari, Hanan Abdulmutal, Mojahed Hadi A. Rudainee

**Affiliations:** 1Internal Medicine Department, College of Medicine, King Khalid University Medical City, Abha 61421, Saudi Arabia; 2Department of Family and Community Medicine, College of Medicine, King Khalid University, Abha 62529, Saudi Arabia; 3Department of Community Medicine, Integral University, Kursi Road, Lucknow 226026, India; 4Neurology Department, Kasr Al Ainy Hospital, Faculty of Medicine, Cairo University, Cairo 4240310, Egypt; 5Neurology Section, Department of Medicine, King Khalid University, Abha 61421, Saudi Arabia; 6Internal Medicine Department, Jazan University, Jazan 45142, Saudi Arabia

**Keywords:** neurological, manifestations, clinical, outcomes, patients, COVID-19, Saudi Arabia

## Abstract

COVID-19 patients also present with rheumatological problems, cardiac problems, and even neurological manifestations. However, the data are still insufficient at present to fill the gaps in our understanding of the neurological presentations of COVID-19. Therefore, the present study was undertaken to reveal the various neurological manifestations of patients with COVID-19 and to find the association between neurological manifestations and the clinical outcome. This cross-sectional study was conducted in Abha, in the Aseer region of the Kingdom of Saudi Arabia, among COVID-19 patients aged 18 years or older who were admitted with the neurological manifestations of COVID-19 to the Aseer Central Hospital and Heart Center Hospital Abha. Non-probability convenient sampling was used. All the information was gathered by the principal investigator using a questionnaire including sociodemographic information, disease characteristics of COVID-19, neurological manifestations, and other complications. Data were analyzed using the Statistical Package for Social Sciences, version 16.0 (SPSS, Inc., Chicago, IL, USA). A total of 55 patients were included in the present study. About half of the patients were admitted to the ICU, and 18 (62.1%) patients died after 1 month of follow-up. Patients aged over 60 years had a 75% mortality rate. About 66.66% of patients with pre-existing neurological disorders died. Statistically significant associations were found between neurological symptoms such as cranial nerve symptoms and a poor outcome. A statistically significant difference was also found between laboratory parameters such as the absolute neutrophil count (ANC), activated partial thromboplastin time (aPTT), total cholesterol (TC), creatinine, urea, and lactate dehydrogenase (LDH) level and the outcome. A statistically significant difference was also found between the use of medications such as antiplatelets, anticoagulants, and statins at the baseline and after a 1-month follow-up. Neurological symptoms and complications are not uncommon among COVID-19 patients. Most of these patients had poor outcomes. Further studies are required to provide more data and knowledge about this issue, including the possible risk factors and the long-term neurological consequences of COVID-19.

## 1. Introduction

The world has faced a series of respiratory tract illnesses, from flu to pneumonia and SARS (severe acute respiratory syndrome), which emerged in China in December 2019 and turned into the coronavirus pandemic, generally called COVID-19. The World Health Organization (WHO) officially declared the COVID-19 pandemic at the beginning of March 2020 [1].

This is the century of illnesses caused by different variants of coronaviruses; since the beginning of this century, three major types of coronavirus-related epidemics have emerged (beta-coronavirus). Surprisingly, two out of three emerged in China. In 2002, the first outbreak of SARS-CoV virus was seen in China (Guangdong Province); a major manifestation of that epidemic comes in the form of the clinical presentation of an acute, very severe type of respiratory syndrome.

In 2012, the second outbreak emerged in the Middle East region, which appeared as MERS CoV in Saudi Arabia, recognized as the Middle East Respiratory Syndrome (MERS). Clinical presentations of MERS CoV are very severe diseases with a very high mortality [2]. The third one is the ongoing pandemic caused by the coronavirus named COVID-19 [3].

The coronavirus causing COVID-19 has impacted the whole world irrespective of age, race, and gender. It has generated huge interest not only from the general public but also from doctors, scientists, paramedics, pharmacological companies, and many other professions all around the globe [4].

COVID-19 has changed the whole world because of its effect on every aspect of daily life, especially psychological and physical effects with its wide variety of clinical manifestations ranging from minor illnesses such as flu, fever, sore throat, abdominal pain, headache, and severe body aches to lethal complications such as breathing difficulties, coagulopathies, SARS, and multi-organ failure. Patients with COVID-19 also present with rheumatological problems, cardiac problems, and even neurological manifestations [5].

Research has shown that patients who present with neurological manifestations have a wide spectrum of symptoms, including those affecting the central and peripheral nervous system which range from mild illness to severe illness of various types. Symptoms include headache, dizziness, impaired consciousness, acute cerebrovascular disease, epilepsy, and peripheral nervous system (PNS)-related manifestations such as hyposmia, anosmia, hypogeusia, ageusia, muscle pain, and Guillain–Barre syndrome (GBS) [6,7,8]. However, many patients with COVID-19 initially present with olfactory and gustatory symptoms, along with respiratory problems which are regarded as early neurological symptoms of SARS-CoV-2 infection, but recover completely [9]. It has also been observed that some patients present with stroke, encephalitis, and other neurological deficits as well as a variable degree of mortality and residual neurological deficits [10]. 

An incomprehensible amount of research has been conducted on COVID-19 to overcome this drastic pandemic. Studies cover different aspects of COVID-19, including not only common clinical manifestations but also other uncommon clinical manifestations, diagnosis, management, treatment, complications, and prevention in terms of vaccine development, acceptability, and issues with their administration as well [11,12,13,14]. There is little concrete evidence-based literature available studying nervous system-related presentations of COVID-19 and their associations with multiple factors. Therefore, the data are still insufficient at present to fill the gap in our understanding of the neurological presentations of COVID-19. Thus, there is a need to conduct further studies to develop a comprehensive understanding of this subject [15].

Therefore, the purpose of the present study is to reveal the various neurological manifestations of patients with COVID-19 in the Aseer region of Saudi Arabia and to find the association of neurological manifestations with the clinical outcome.

## 2. Material and Methods

### 2.1. Study Area and Population

This study was conducted in Abha, in the Aseer region of the Kingdom of Saudi Arabia. Abha is one of the hill stations of the Sarawat Mountains. A cross-sectional study was conducted among the COVID-19 patients who were admitted with neurological manifestations of COVID-19 to the Aseer Central Hospital and Heart Center Hospital Abha. These are the only two government hospitals in the city which were providing care for patients during the lockdown due to the COVID-19 pandemic. During the lockdown period due to of COVID-19, all outpatient facilities were suspended and patients were managed either through the emergency unit or online consultation. Only patients with severe problems were admitted to hospitals for indoor care, whereas some patients with more severe symptoms were admitted to the ICU for care. Data were collected over a 12-month period starting from the end of April 2020 to March 2021. 

### 2.2. Inclusion and Exclusion Criteria

The COVID-19 pandemic was an exceptional situation that the whole world faced at the same time. Lots of unusual presentations were identified and neurological manifestations were one of them. All patients aged 18 years or above that presented with neurological manifestations and were admitted to the selected hospitals were included in the study. All males or females of any ethnicity with a confirmed COVID-19 infection and presenting acute neurological signs and symptoms were included. All the information was gathered by the principal investigator; the patients’ confidentiality and anonymity were maintained throughout the data collection. All the patients with a pre-existing neurological disease with a new onset of signs and symptoms were also included. Neurological symptoms were included, such as headache, confusion, lack of sleep, speech changes, visual changes, weakness, convulsions, sensory changes, unsteadiness, dizziness, myalgia, and back pain, whereas the neurological signs included those such as coma, cognitive or behavioral changes, signs of meningeal irritation, aphasia/dysarthria, ataxia, insomnia, focal neurological deficit (hemiparesis/hemisensory loss) and cranial neuropathy (oculomotor signs; diplopia, nystagmus, blindness), and other cranial neuropathy (facial palsy and dysphagia). Complications were included, such as death and long-term neurological deficits. A postmortem was not conducted in any of the patients to confirm the cause of death. 

All patients under 18 years of age were excluded. Pregnant women were also excluded from this research. 

### 2.3. Sample Size and Sampling Technique

Non-probability convenient sampling was done and all the patients who fulfilled the inclusion criteria were recruited accordingly in the study. A total of 55 patients were selected for the study.

### 2.4. Ethical Approval and Data Collection

The study was approved by the ethics and IRB committee of Aseer Central Hospital. All the information was gathered by the principal investigator by using a questionnaire, including the patients’ sociodemographic information, disease characteristics of COVID-19, neurological manifestations, and other complications. 

### 2.5. Statistical Analysis

The collected data were coded and entered into an Excel software (Microsoft Office Excel 2010) database. Data were analyzed using the Statistical Package for Social Sciences, version 16.0 (SPSS, Inc., Chicago, IL, USA). Information related to radiological reporting, the onset of neurological symptoms, and the smoking status of patients was presented in pie and clustered bar graphs. Pearson’s chi-square test was used to assess the association between patient outcomes and demographic characteristics, co-morbid conditions, and neurological symptoms. An analysis of variance test was used to find the mean differences between the patient outcomes and laboratory parameters. Wilcoxon’s signed-ranks test was used for a comparison of medications at the baseline and after one month. A *p*-value of less than 0.05 was regarded as statistically significant. Multivariate logistic regression analysis was also used.

## 3. Results

Table 1 shows the association of patients’ outcomes with demographic characteristics and co-morbid conditions. A total of 55 patients were included in the present study. More than 3/4 of the patients were Saudi nationals. The ratio of male and female patients was 7:4. About half of the total patients were admitted to the ICU, in which 18 (62.1%) patients died after 1 month of follow-up. Patients aged less than sixty years had a higher percentage of complete recovery. Patients aged more than 60 years had 75% mortality. Approximately half of the patients were suffering from diabetes mellitus and hypertension. After a 1-month follow-up, all the patients died with co-morbidity, such as congenital heart disease, chronic pulmonary disease, liver disease, or on immunosuppressive therapy, asthma, and rheumatological autoimmune disease. The majority of patients had pre-existing neurological disease (21.8%), followed by heart disease (18.2%) and endocrinological disease (9.1%). Almost 60% of diabetic patients also died of COVID-19-associated neurological deficits. About 66.66% of patients with pre-existing neurological disorders died. 

Table 2 shows the association of patient outcomes with neurological symptoms such as a change in the level of consciousness (70.9%), headache (70.9%), weakness (43.6%), dysarthria (40.0%), seizures (29.1%), cranial nerves symptoms (25.4%), aphasia (23.6%), gait unsuitability (14.5%), myalgia (3.6%), autonomic symptoms (5.5%), sensory symptoms (3.6%), abnormal movement (1.8%), and ataxia (1.8%) among the total amount of patients. Cranial nerve involvement was found among 25.45% of patients. Statistically significant associations were found between neurological symptoms such as cranial nerve symptoms and weakness with the outcome. 

Table 3 shows the difference between the laboratory parameters and the patient outcome using an analysis of variance. The average platelets count, ALC, D dimer, LDL-C, and TC were higher in patients with residual neurological deficits. The average Hb level was approximately the same in all categories of the outcome after a 1-month follow-up. The average value of WBC, ANC, PT, aPTT, INR, TG, creatinine, urea, LDH level, vitamin D, ferritin, HBA1C, and CK total reached the maximum in dead patients after a 1-month follow-up. Statistically, a significant difference was found between laboratory parameters such as ANC, aPTT, TC, creatinine, urea, and the LDH level and the outcome.

Table 4 shows the patient’s medication status at the baseline and after 1 month. At the baseline, 37 (67.3%) patients were not taking any type of anti-platelets. However, after one month, it increased to 46 (83.6%). A severe decline was found in a patient taking direct oral anticoagulants (DOACs) from the baseline to after 1 month. Statistically, a significant difference was found between the type of medication taken at the baseline and after a 1-month follow-up. 

Table 5 shows that no statistically significant associations were found between subjects having less than and more than two co-morbidities conditions and mortality, recovery, and a neurological deficit. Out of the total subjects, 40% had more than two co-morbidities.

Table 6 includes those variables undergoing a significant univariate test at some random level chosen as a variable for multinomial logistic regression to estimate the parameters after 1 month of follow-up. Each factor involved an assessment with the reference group, which is characterized as the patient dying after a one-month follow-up. For those belonging to the age group under 60 years of age, relative to more than 60 years, the complete recovery of patients relative to their mortality, as well as the residual neurological deficits of patients relative to their mortality, would be expected to increase by multiple factors after 1-month follow-up patients.

Figure 1 represents the onset of neurological symptoms among COVID-19 patients. The onset of neurological symptoms in the majority of patients (45%) was shown to be between 3 days and 10 days and in 26% of patients, symptoms developed between 10 days and 14 days after their COVID-19 diagnosis. Only 12% of patients developed neurological symptoms after 14 days of COVID-19 diagnosis. 

Figure 2 distinguished the patients into three groups: people who had never smoked, former smokers, and current smokers. This depicts that 100% of patients who died were current smokers, followed by 75% of patients who died who were former smokers. Twenty-six percent of complete recoveries were observed in those patients who had never smoked. 

## 4. Discussion

In our study, we identified unusual signs and symptoms of coronavirus infection during the COVID-19 pandemic. Many patients presented with signs and symptoms related to the central and peripheral nervous systems. These findings suggested that SARS-CoV-2 virus does not confine to a vast variety of respiratory illnesses [16].

In our study, female patients made up about 36% of the cohort, which was similar to the findings of a study conducted in Spain, where 38% of females were enrolled in the study with neurological manifestations [17]. The mortality rate was around 62% among our patients after a one-month follow-up; however, the death rate was found to be of a variable degree in different settings. A study conducted in the United States by Salahuddin H et al. reported 37.4% mortality among patients with major neurological manifestations [18]. As shown from previous studies, the viral infection caused by SARS-CoV-2 plays a potential role in developing cerebrovascular events in several patients [19]. It is also evident from the literature that COVID-19 affected the older population disproportionately; there was a higher probability of hospitalizations, complications, and mortality in this age category [20]. In our study, we found that patients under sixty years of age had a higher percentage of complete recovery and those aged older than 60 years had a 75% mortality rate. This age predilection for neurological manifestations and complications related to COVID-19 was supported by De Biase S in his study as well [21]. It is has also been stated by previous studies that pre-existing comorbid conditions are correlated with disease severity, mortality, and complications as well. The most prevalent comorbid conditions were cardiometabolic diseases such as hypertension, diabetes, and coronary artery disease among patients with COVID-19. In our study, approximately half of the patients were suffering from diabetes and hypertension as comorbid conditions; after a 1-month follow-up, all the patients who died were those with co-morbidity such as congenital heart diseases, chronic pulmonary disease, liver disease, on immunosuppressive therapy, asthma, and rheumatological autoimmune diseases. This is also supported by the literature in terms of comorbid diseases being directly associated with disease severity, complications, and mortality among COVID-19 patients [22]. Among our patients, we found the following frequencies of neurological symptoms among the total number of patients in our study: change in the level of consciousness (LOC) 39 (70.9%), headache 39 (70.9%), weakness 24 (43.6%), dysarthria 22 (40.0%), seizures 16 (29.1%), cranial nerves symptoms 14 (25.4%), aphasia 13 (23.6%), gait unsuitability 8 (14.5%), myalgia 2 (3.6%), autonomic symptoms 3 (5.5%), sensory symptoms 2 (3.6%), abnormal movement 1 (1.8%), and ataxia 1 (1.8%). In other studies, they found different frequencies of neurological symptoms [23,24]. A study conducted by Misra S et al. showed the prevalence of the most common neurological symptoms in their patients, which included fatigue (32%), myalgia (20%), taste impairment (21%), smell impairment (19%), and headache (13%). For patients older than 60 years of age, the prevalence of acute confusion/delirium was 34% among patients with COVID-19. The presence of any of the neurologic manifestations in age > 60 years was associated with a high mortality [25]. The involvement of cranial nerves is being increasingly identified among patients with COVID-19. In our study, among neurological deficits, 25.45% of patients had cranial nerve involvement. A high percentage of the literature has also been dedicated to cranial nerve involvement and many other types of research also support the evidence of cranial nerve involvement with COVID-19 infection [26,27,28,29,30]. Almost 60% of diabetes and 70% of patients with heart diseases died from COVID-19-associated neurological deficits in our study. This is also supported by the literature in the way that the majority of patients with chronic diseases such as diabetes, hypertension, heart disease, and pulmonary diseases had a high mortality rate compared to the patients with no co-morbidity [31,32,33,34]. Among our patients, 66.66% of patients died from pre-existing neurological disorders. A study conducted by García-Azorín D et al. also supported that the presence of chronic neurological disorders was an independent predictor of death (HR 2.129, 95% CI: 1.382–3.280) but not a severer COVID-19 disease (OR: 1.75, 95% CI: 0.970–3.158) [35].

The results of our study showed that after a 1-month follow-up, all patients died with neurological symptoms such as myalgia, abnormal movement, sensory symptoms, and ataxia. Although these were the most reported symptoms, the mortality rate varies, regarding dissimilar symptoms of neurological deficits [7,36]. For cases of COVID-19, some abnormal laboratory test results may predict the progress of the patient, clinical course, outcome, and mortality as well [37,38]. In this study, we found a statistically significant difference between the laboratory parameters, such as ANC, aPTT, TC, creatinine, urea, and LDH level, and the outcomes after a 1-month follow-up. Ruan Q reported in his study, which was conducted in Wuhan, that regarding the laboratory results, there were significant differences in white blood cell counts, absolute values of lymphocytes, platelets, albumin, total bilirubin, blood urea nitrogen, blood creatinine, myoglobin, cardiac troponin, C-reactive protein (CRP), and interleukin-6 (IL-6) among their patients [39]. Among our patients, the average values of WBC, ANC, PT, aPTT, INR, TG, creatinine, urea, LDH level, vitamin D, ferritin, HBA1C, and CK total reached the maximum in dead patients after a 1-month follow-up. High levels of glycosylated hemoglobin, which is considered a surrogate marker for long-term blood sugar control among diabetic patients, have also been linked to causing inflammation, hypercoagulation, and high mortality among patients [40]. Laboratory findings commonly associated with worse outcomes include raised D-dimer levels, C-reactive protein (CRP), LDH, and high-sensitivity cardiac troponin I [41,42]. Antiplatelets, anticoagulants, and statins were being administered to COVID-19 patients as anti-inflammatory medications. However, 37 (67.3%) of our patients were not taking any anti-platelets, 21 (38.2%) were not on any anticoagulants, and 41 (74.5%) were not on any anti-dyslipidemia medications upon their admission.

Antiplatelet agents have a systemic antithrombotic action, which is associated with decreased mortality among COVID-19 patients. Regarding anti-platelet therapy, it may be related to aspirin’s antiviral effects and anti-inflammatory effects, which are not found in other antiplatelet agents [43]. The literature also suggests the beneficial role of anticoagulants in decreasing mortality [44]. However, a meta-analysis conducted by Hariyanto TI reported that the use of statins did not improve the severity outcome (OR 1.64 95% CI 0.51–5.23) nor the mortality rate from COVID-19 infection (OR 0.78 95% CI 0.50–1.21) [45].

Our study has a few limitations. The sample used to conduct this research is small and may not be representative of the general population of hospitalized patients. However, the data presented in our study are of a one-year duration of the COVID-19 pandemic. Twenty-nine patients were only admitted to the ICU with neurological deficits during this period. Aseer Central Hospital was the only government hospital in Abha city that was providing health services during this period. The cross-sectional design cannot confirm the causality of the relationship between the compared variables. We decided to conduct a cross-sectional study as, practically, it was not feasible in the COVID-19 era to conduct a case–control or cohort study. However, in the future, we hope to acquire all the required resources to conduct case–control or cohort and multicentric/nationwide studies. The strength of our study is that in the Kingdom of Saudi Arabia, this study is the first of its own kind which presents the neurological manifestations and clinical outcomes of COVID-19 patients. This topic has been extensively studied in other parts of the world, but associations with mortality and a residual neurological deficit are not yet conclusive. We included 55 patients from the hospital data and demonstrated the mortality rate under different conditions. We also listed many patient laboratory parameters, which can be an important reference in clinical practice.

## 5. Conclusions

Neurological symptoms and complications are not uncommon among COVID-19 patients. Most of these patients had poor outcomes. Further studies are required to provide more data and knowledge about this issue, including the possible risk factors and the long-term neurological consequences of COVID-19.

## Figures and Tables

**Figure 1 ijerph-20-03848-f001:**
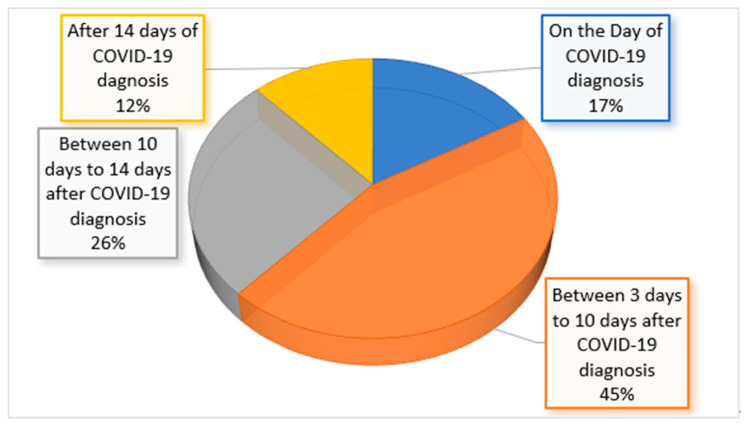
The onset of neurological symptoms among COVID-19 patients.

**Figure 2 ijerph-20-03848-f002:**
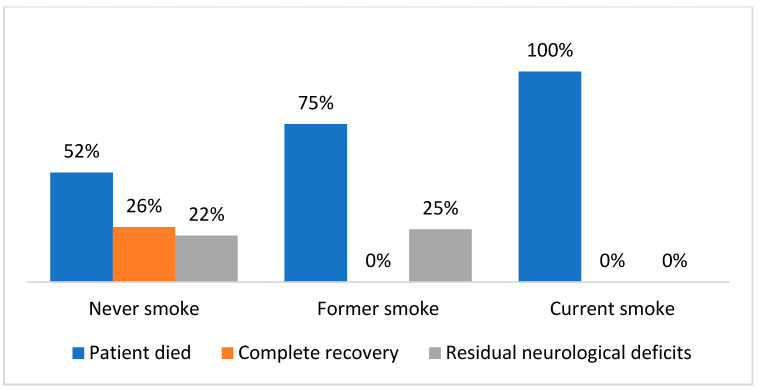
Percent distribution of smoking status of patients.

**Table 1 ijerph-20-03848-t001:** Relation of patient’s outcome with demographic characteristics and co-morbid conditions.

Demographics Characteristics	Category	Patient Outcome after 1-Month Follow-Up?			Total	*p*-Value
		Mortality (n = 30)	Complete Recovery (n = 13)	Residual Neurological Deficits (n = 12)		
ICU patients	No	12 (46)	7 (26.9)	7 (26.9)	26 (100)	0.00 *
	Yes	18 (62.1)	6 (20.7)	5 (17.2)	29 (100)	
Age group (in years)	≤60	6 (26.1)	9 (39.1)	8 (34.8)	23 (100)	0.00 *
	>60	24 (75)	4 (12.5)	4 (12.5)	32 (100)	
Gender	Male	20 (57.1)	6 (17.1)	9 (25.7)	35 (100)	0.29 *
	Female	10 (50)	7 (35)	3 (15)	20 (100)	
Nationality	Saudi	25 (59.5)	11 (26.2)	6 (14.3)	42 (100)	0.05 *
	Non-Saudi	5 (38.5)	2 (15.4)	6 (46.2)	13 (100)	
Heart disease	No	23 (51.1)	12 (26.7)	10 (22.2)	45 (100)	0.46
	Yes	7 (70)	1 (10)	2 (20)	10 (100)	
Congenital heart disease	No	29 (53.7)	13 (24.1)	12 (22.2)	54 (100)	0.65
	Yes	1 (100)	0 (0.0)	0 (0.0)	1 (100)	
Chronic pulmonary disease	No	27 (51.9)	13 (25)	12 (23.1)	52 (100)	0.26
	Yes	3 (100)	0 (0.0)	0 (0.0)	3 (100)	
Renal disease	No	27 (52.9)	13 (25.5)	11 (21.6)	51 (100)	0.50
	Yes	3 (75)	0 (0.0)	1 (25)	4 (100)	
Liver disease	No	29 (53.7)	13 (24.1)	12 (22.2)	54 (100)	0.65
	Yes	1 (100)	0 (0.0)	0 (0.0)	1 (100)	
Neurological diseases ∞	No	22 (51.2)	11 (25.6)	10 (23.3)	43 (100)	0.63
	Yes	8 (66.7)	2 (16.7)	2 (16.7)	12 (100)	
Endocrinological diseases	No	27 (54)	11 (22)	12 (24)	50 (100)	0.40
	Yes	3 (60)	2 (40)	0 (0.0)	5 (100)	
Immunosuppressive therapy	No	27 (51.9)	13 (25)	12 (23.1)	52 (100)	0.26
	Yes	3 (100)	0 (0.0)	0 (0.0)	3 (100)	
Diabetes mellitus	No	12 (48)	7 (28)	6 (24)	25 (100)	0.68
	With complications	14 (56)	6 (24)	5 (20)	25 (100)	
	Without complications	4 (80)	0 (0.0)	1 (20)	5 (100)	
Hypertension	No	13(48.1)	7 (25.9)	7 (25.9)	27 (100)	0.62
	Yes	17 (60.7)	6 (21.4)	5 (17.9)	28 (100)	
Asthma	No	28 (52.8)	13 (24.5)	12 (22.6)	53 (100)	0.42
	Yes	2 (100)	0 (0.0)	0 (0.0)	2 (100)	
Rheumatological autoimmune disease	No	28 (52.8)	13 (24.5)	12 (22.6)	53 (100)	0.42
	Yes	2 (100)	0 (0.0)	0 (0.0)	2 (100)	

* Chi-square, #ANOVA. ∞ Neurological disease (Guillan–Barre syndrome, left eye and angle of mouth twitching with colonic movements, ischemic stroke, myasthenia gravis, previous intra cranial hemorrhage, previous ischemic stroke, previous meningoencephalitis, dementia, psychiatric disorder, two previous ischemic strokes, Parkinson’s disease), endocrinological disease (hypothyroidism), heart diseases (atrial fibrillation, ischemic heart disease, heart failure), respiratory disease (increased pulmonary artery pressure, rheumatoid lung disease), renal disease (CKD, ESRD on HD, ESRD on dialysis).

**Table 2 ijerph-20-03848-t002:** Relation of patient outcomes with the neurological symptoms.

		Patient Outcome after 1-Month Follow-Up?			Total	*p*-Value *
		Mortality (n = 30)	Complete Recovery (n = 13)	Residual Neurological Deficits (n = 12)		
Change LOC	No	11 (68.8)	1 (6.2)	4 (25)	16 (100)	0.14
	Yes	19 (48.7)	12 (30.8)	8 (20.5)	39 (100)	
Dysarthria	No	20 (60.6)	8 (24.2)	5 (15.2)	33 (100)	0.32
	Yes	10 (45.5)	5 (22.7)	7 (31.8)	22 (100)	
Aphasia	No	22 (52.4)	11 (26.2)	9 (21.4)	42 (100)	0.72
	Yes	8 (61.5)	2 (15.4)	3 (23.1)	13 (100)	
Cranial nerves symptoms	No	22 (53.7)	13 (31.7)	6 (14.6)	41 (100)	0.01
	Yes	8 (57.1)	0 (.0)	6 (42.9)	14 (100)	
Weakness	No	17 (54.8)	11 (35.5)	3 (9.7)	31 (100)	0.01
	Yes	13 (54.2)	2 (8.3)	9 (37.5)	24 (100)	
Myalgia	No	28 (52.8)	13 (24.5)	12 (22.6)	53 (100)	0.42
	Yes	2 (100)	0 (.0)	0 (.0)	2 (100)	
Abnormal movement	No	29 (53.7)	13 (24.1)	12 (22.2)	54 (100)	0.65
	Yes	1 (100)	0 (.0)	0 (.0)	1 (100)	
Sensory symptoms	No	28 (52.8)	13 (24.5)	12 (22.6)	53 (100)	0.42
	Yes	2 (100)	0 (.0)	0 (.0)	2 (100)	
Ataxia	No	29 (53.7)	13 (24.1)	12 (22.2)	54 (100)	0.65
	Yes	1 (100)	0 (.0)	0 (.0)	1 (100)	
Gait unsuitability	No	26 (55.3)	13 (27.7)	8 (17)	47 (100)	0.05
	Yes	4 (50)	0 (.0)	4 (50)	8 (100)	
Autonomic symptoms	No	28 (53.8)	13 (25)	11 (21.2)	52 (100)	0.59
	Yes	2 (66.7)	0 (.0)	1 (33.3)	3 (100)	
Seizures	No	21 (53.8)	8 (20.5)	10 (25.6)	39 (100)	0.48
	Yes	9 (56.2)	5 (31.2)	2 (12.5)	16 (100)	
Headache	No	12 (75)	3 (18.8)	1 (6.2)	16 (100)	0.10
	Yes	18 (46.2)	10 (25.6)	11 (28.2)	39 (100)	

* Chi-square.

**Table 3 ijerph-20-03848-t003:** Relation of laboratory parameters with the patient outcome by using ANOVA.

Parameter	Patient Outcome after 1-Month Follow-Up?	Mean	Std. Deviation	95% Confidence Interval for Mean		F	*p*-Value *
				Lower Bound	Upper Bound		
Hemoglobin (Hb) (12–16) g/dl	Mortality (n = 30)	13.29	2.052	12.53	14.06	0.006	0.994
	Complete recovery (n = 13)	13.34	2.325	11.93	14.74		
	Residual neurological deficits (n = 12)	13.38	2.844	11.57	15.18		
	Total (n = 55)	13.32	2.261	12.71	13.93		
Platelets (130–400) × 10^9^/L)	Mortality (n = 30)	193.61	86.646	161.26	225.97	2.814	0.069
	Complete recovery (n = 13)	251.54	119.277	179.46	323.62		
	Residual neurological deficits (n = 12)	260.24	96.197	199.12	321.36		
	Total (n = 55)	221.84	100.305	194.72	248.96		
White blood cell count (WBC) (4.5 to 11.0 × 10^9^/L)	Mortality (n = 30)	10.70	4.837	8.90	12.51	3.114	0.053
	Complete recovery (n = 13)	7.42	4.167	4.90	9.94		
	Residual neurological deficits (n = 12)	7.79	4.445	4.97	10.62		
	Total (n = 55)	9.29	4.785	8.00	10.59		
Absolute neutrophil count (ANC) 1.8 to 7.5 × 10^9^/L	Mortality (n = 30)	9.26	4.671	7.52	11.01	5.326	0.008
	Complete recovery (n = 13)	5.41	2.284	4.03	6.79		
	Residual neurological deficits (n = 12)	6.32	3.134	4.33	8.31		
	Total (n = 55)	7.71	4.234	6.56	8.85		
Absolute lymphocytic count (ALC) (1–4.8 × 10^9^/L)	Mortality (n = 30)	1.32	0.888	0.99	1.65	0.085	0.919
	Complete recovery (n = 13)	1.22	0.623	0.84	1.60		
	Residual neurological deficits (n = 12)	1.34	0.725	0.88	1.80		
	Total (n = 55)	1.30	0.787	1.09	1.51		
Prothrombin time (PT) (9.0–14.0) second	Mortality (n = 30)	13.05	2.641	12.06	14.03	1.862	0.166
	Complete recovery (n = 13)	11.72	1.980	10.53	12.92		
	Residual neurological deficits (n = 12)	12.03	1.404	11.14	12.93		
	Total (n = 55)	12.51	2.319	11.89	13.14		
Partial thromboplastin time (aPTT) (30.0–40.0) second	Mortality (n = 30)	34.12	8.473	30.95	37.28	4.891	0.011
	Complete recovery (n = 13)	28.04	4.146	25.54	30.55		
	Residual neurological deficits (n = 12)	28.23	5.286	24.87	31.59		
	Total (n = 55)	31.40	7.557	29.35	33.44		
International normalized ratio (INR) (1–1.5)	Mortality (n = 30)	2.40	5.324	0.37	4.42	0.780	0.464
	Complete recovery (n = 13)	1.02	0.121	0.94	1.09		
	Residual neurological deficits (n = 12)	1.09	0.244	0.93	1.24		
	Total (n = 55)	1.77	3.930	0.70	2.85		
D-dimer (<500 ng/mL)	Mortality (n = 30)	4.45	0.719	4.18	4.71	1.631	0.206
	Complete recovery (n = 13)	3.48	1.651	2.49	4.48		
	Residual neurological deficits (n = 12)	4.74	3.525	2.50	6.98		
	Total (n = 55)	4.28	1.905	3.77	4.80		
C-reactive proteins (CRP) (0.3–1.0 mg/dL)	Mortality (n = 30)	0.37	0.490	0.18	0.55	0.650	0.526
	Complete recovery (n = 13)	0.54	0.519	0.22	0.85		
	Residual neurological deficits (n = 12)	0.50	0.522	0.17	0.83		
	Total (n = 55)	0.44	0.501	0.30	0.57		
Low-density lipoproteins (LDL-C) (<100 mg/dL).	Mortality (n = 30)	61.08	25.088	51.72	70.45	5.752	0.006
	Complete recovery (n = 13)	75.44	11.795	68.31	82.57		
	Residual neurological deficits (n = 12)	86.76	26.688	69.80	103.72		
	Total (n = 55)	70.08	25.054	63.31	76.85		
High-density lipoproteins (HDL-C) (60 and above mg/dL).	Mortality (n = 30)	33.20	9.914	29.50	36.90	0.531	0.591
	Complete recovery (n = 13)	36.05	8.745	30.77	41.34		
	Residual neurological deficits (n = 12)	33.23	3.817	30.81	35.66		
	Total (n = 55)	33.88	8.616	31.55	36.21		
Total cholesterol (TC) (<200 mg/dL).	Mortality (n = 30)	139.19	48.918	120.92	157.46	5.428	0.007
	Complete recovery (n = 13)	165.43	21.711	152.31	178.55		
	Residual neurological deficits (n = 12)	183.78	37.231	160.12	207.44		
	Total (n = 55)	155.12	44.959	142.97	167.27		
Triglycerides (TG) (<150 mg/dL).	Mortality (n = 30)	190.67	111.086	149.19	232.15	0.498	0.611
	Complete recovery (n = 13)	189.15	112.117	121.40	256.91		
	Residual neurological deficits (n = 12)	157.17	49.801	125.52	188.81		
	Total (n = 55)	183.00	100.577	155.81	210.19		
Creatinine (Cr) (0.7–1.2 mg/dL)	Mortality (n = 30)	1.45	0.661	1.20	1.70	5.225	0.009
	Complete recovery (n = 13)	0.94	0.451	0.66	1.21		
	Residual neurological deficits (n = 12)	0.90	0.605	0.52	1.28		
	Total (n = 55)	1.21	0.652	1.03	1.38		
Urea (7–30) mg/dL	Mortality (n = 30)	86.29	56.848	65.06	107.51	4.674	0.014
	Complete recovery (n = 13)	56.10	39.019	32.52	79.68		
	Residual neurological deficits (n = 12)	38.67	32.075	18.29	59.05		
	Total (n = 55)	68.76	51.904	54.73	82.79		
Lactate dehydrogenase (LDH) (120–227 U/L)	Mortality (n = 30)	611.24	216.745	530.30	692.17	8.490	0.001
	Complete recovery (n = 13)	364.65	153.747	271.74	457.56		
	Residual neurological deficits (n = 12)	479.23	111.893	408.14	550.32		
	Total (n = 55)	524.15	209.329	467.56	580.74		
Vitamin D (≥30 ng/mL)	Mortality (n = 30)	29.16	6.611	26.69	31.63	2.918	0.063
	Complete recovery (n = 13)	26.58	3.178	24.66	28.50		
	Residual neurological deficits (n = 12)	24.16	7.650	19.29	29.02		
	Total (n = 55)	27.46	6.470	25.71	29.21		
Ferritin (≥30.0 ng/mL)	Mortality (n = 30)	1446.63	667.210	1197.49	1695.77	1.899	0.160
	Complete recovery (n = 13)	1270.74	407.773	1024.32	1517.15		
	Residual neurological deficits (n = 12)	1054.78	577.330	687.96	1421.60		
	Total (n = 55)	1319.56	607.489	1155.33	1483.79		
Glycosylated hemoglobin (HbA1c) (less than 7.0%)	Mortality (n = 30)	8.27	0.910	7.93	8.61	2.513	0.091
	Complete recovery (n = 13)	8.03	0.701	7.61	8.46		
	Residual neurological deficits (n = 12)	7.62	0.817	7.10	8.14		
	Total (n = 55)	8.07	0.870	7.83	8.30		
Creatine kinase (CK) (26–192) U/L	Mortality (n = 30)	785.62	914.909	443.99	1127.26	0.899	0.413
	Complete recovery (n = 13)	508.94	294.162	331.18	686.70		
	Residual neurological deficits (n = 12)	446.83	1063.725	−229.03	1122.69		
	Total (n = 55)	646.31	850.542	416.37	876.24		

* ANOVA.

**Table 4 ijerph-20-03848-t004:** Comparison of medications at baseline and after one month.

Medications		Baseline		After a 1-Month Follow-Up?		*p*-Value *
		n	%	n	%	
Antiplatelets	No	37	67.3	46	83.6	0.03
	Aspirin	13	23.6	7	12.7	
	Clopidogrel	3	5.5	1	1.8	
	Both	2	3.6	1	1.8	
Anticoagulants	No	21	38.2	48	87.3	0.00
	Warfarin	0	0	2	3.6	
	Direct oral anticoagulants (DOACs)	34	61.8	5	9.1	
Statins	No	41	74.5	49	89.1	0.06
	Atorvastatin	11	20.0	5	9.1	
	Rosuvastatin	2	3.6	0	0	
	Simvastatin	1	1.8	1	1.8	

* Wilcoxon signed-ranks.

**Table 5 ijerph-20-03848-t005:** Association between subjects having co-morbidities and mortality, recovery, and neurological deficit.

Number of Comorbid Conditions	Patient Outcome after 1-Month Follow-Up			Total	*p*-Value
	Mortality (n = 30)	Complete Recovery (n = 13)	Residual Neurological Deficits (n = 12)		
≤2	15 (45.5%)	10 (30.3%)	8 (24.2%)	33 (100.0%)	0.22
>2	15 (68.2%)	3 (13.6%)	4 (18.2%)	22 (100.0%)	

**Table 6 ijerph-20-03848-t006:** Multinomial logistic regression to estimate the association of parameters.

Parameters	One-Month Follow-Up		B	Std. Error	Wald	*p*-Value	Odds Ratio
Complete recovery after 1-month follow-up		Intercept	292.74	12.27	568.86	0.00	
	Investigations	ANC	−10.16	1.14	79.05	0.00	0.00
		aPTT	−4.93	0.19	628.73	0.00	0.01
		LDC_C	0.25	0.08	8.71	0.00	1.29
		TC	1.31	0.03	1482.00	0.00	3.72
		Creatinine	−8.53	4.47	3.63	0.06	0.00
		Urea	−0.23	0.08	8.20	0.00	0.79
		LDH Level	−0.15	0.01	100.07	0.00	0.86
		Yes	−278.45	7.97	1218.00	0.00	0.00
	Age group (in years)	≤60	31.49	4.50	48.81	0.00	4.75 × 10^13^
	Country	Saudi	84.63	2.66	1010.00	0.00	5.70 × 10^53^
	Cranial nerves symptoms	Yes	−58.531	6.07	92.78	0.00	0.00
	Motor function neurological assessment	Weakness	−87.41	5.62	241.96	0.00	0.00
		Gait unsuitability	−162.61	0.00			0.00
Residual neurological deficits after 1-month follow-up		Intercept	616.01	12.34	2492.00	0.00	
	Investigations	ANC	−5.66	0.86	43.22	0.00	0.00
		aPTT	−15.69	0.24	4001.00	0.00	0.00
		LDC_C	0.58	0.03	245.51	0.00	1.80
		TC	−0.03	0.03	0.74	0.39	0.97
		Creatinine	−87.47	3.20	747.41	0.00	0.00
		Urea	0.85	0.06	152.63	0.00	2.36
		LDH level	0.0	0.01	36.79	0.00	1.09
		Yes	−215.20	4.36	2433.00	0.00	0.00
	Age group (in years)	≤60	57.51	2.63	476.51	0.00	9.48 × 10^23^
	Country	Saudi	−2.22	2.61	0.73	0.39	0.11
	Cranial nerves symptoms	Yes	43.44	2.79	242.41	0.00	7.40 × 10^18^
	Motor function neurological assessment	Weakness present	2.35	4.50	0.27	0.60	10.54
		Gait unsuitability	37.52	2.82	177.10	0.00	1.98 × 10^18^

Note: The reference category is: no.

## Data Availability

Not applicable.

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
