# Peer review of "Neurological Manifestations and Clinical Outcomes of Patients with COVID-19 in the Aseer Region, Saudi Arabia"

_ijerph, 2023, doi:10.3390/ijerph20053848_

Round 1
Reviewer 1 Report
In this manuscript, authors intend to provide insight on the neurological manifestations of COVID-19 and their association with clinical outcomes. I appreciate the effort to extract the maximum information from such a small sample of patients and the ability of the authors to explain their methods and results in extensive detail. I have found no serious flaws, incongruences, or errors in the conduction of the proposed research. However, I believe that the evidence provided is of little clinical value. The main reasons for this statement, which I hope the authors can address in the present or future works, are:
1. The sample used to conduct this research is very small and not representative of the general population of hospitalised patients (i.e., the number of patients admitted to the ICU is dramatically high).
2. If authors wanted to address the role of neurological manifestations in clinical outcomes, maybe they could have compared their recruited cohort with another one where these symptoms are absent.
3. Authors based all their assumptions and conclusions in simple bivariable comparisons, when the proposed goal would have required more advanced statistical methods (such as multivariable logistic, linear or Cox regression) to account for possible confounding factors of unfavourable outcome and to identify strong predictors.
4. The topic of the present manuscript has already been studied extensively, and better quality evidence exists at present in the literature.
For all the above mentioned reasons, in my humble opinion, the novelty and reliability of this manuscript remain questionable.
Author Response
Dear Reviewer, Thank you for the suggestions. Please see our reply.
- The sample used to conduct this research is very small and not representative of the general population of hospitalised patients (i.e., the number of patients admitted to the ICU is dramatically high).
Reply: As pointed out by the respected reviewer the sample size may be a limitation of our study. We have included this as a limitation. However the data presented in our study is of one year duration in COVID pandemic situation. Twenty nine patients were only admitted in the ICU with neurological deficits during this period. Aseer Central hospital was the only government hospital in Abha city which was providing health services during COVID-19 pandemic lockdown. The strength of our study is that we performed this study on neurological manifestations and clinical outcomes of patients with COVID-19 based on the data collected in the hospital. We included 55 patients and demonstrated the mortality rate under different conditions. We also listed many patient laboratory parameters, which can be an important reference in clinical practice. (Page 13, Please see lines 327-341)
- If authors wanted to address the role of neurological manifestations in clinical outcomes, maybe they could have compared their recruited cohort with another one where these symptoms are absent.
Reply: As pointed out by the respected reviewer the study design may be a limitation of our study. We have included this as a limitation. During lock down period because of COVID-19 all outpatient facilities were suspended. The patients were managed either in emergency room or admitted in wards, only sick patients were admitted in ICU care. Health system was overburdened; in that situation it was not possible for us to compare our cohort of patients with other patients with no such symptoms.
We preferred to conduct a cross sectional study; as practically it was not feasible with in COVID era, an emergency situation to conduct case control or cohort study. However in future we hope to have all the required resources to do cohort and multicentric /nationwide studies. (Page 13, Please see lines 327-341)
- Authors based all their assumptions and conclusions in simple bivariable comparisons, when the proposed goal would have required more advanced statistical methods (such as multivariable logistic, linear or Cox regression) to account for possible confounding factors of unfavourable outcome and to identify strong predictors.
Reply: As suggested by the respected reviewer, Table 6 has been added with logistic regression to account for possible confounding factors of unfavorable outcome and to identify strong predictors. (Please see page 10, lines 214-219)
- The topic of the present manuscript has already been studied extensively, and better quality evidence exists at present in the literature.
Reply: As pointed by the respected reviewer, this topic has been extensively studied in other parts of the world but associations with mortality and residual neurological deficit is not conclusive yet. In the Kingdom of Saudi Arabia, this study is first of its own kind, presenting the neurological manifestations and clinical outcomes COVID-19 patients. The study conclusion may help in strategic planning and prevention and control of the pandemic situation. (Please see page 13, lines 337-343)
- For all the above mentioned reasons, in my humble opinion, the novelty and reliability of this manuscript remain questionable.
Reply: The COVID-19 pandemic was not a normal period. The global emergency caused huge number of deaths. Medical conditions were handled with great difficulties during lockdown, all the hospitals were overloaded and many doctors lost their lives also. In such circumstances observing for different and unusual manifestations other than respiratory symptoms of SARS CoV2 virus was not an easy task. We studied the neurological manifestations and clinical outcomes of patients with COVID-19, based on a government hospital data. We included 55 patients and demonstrated the mortality rate under different conditions. We also listed many patient laboratory parameters, which can be an important reference in clinical practice.
Reviewer 2 Report
The authors performed a fine study on neurological manifestations and clinical outcomes of patients with COVID-19 based on the data collected in their hospital. They included 55 patients and demonstrated the mortality rate under different conditions. They also listed many patient laboratory parameters, which can be an important reference in clinical practice. Here are my comments.
Table 1. According to their data, many patients had more than two comorbidities. It would be necessary to make a statistic on this issue.
How were the causes of death of patients diagnosed? Did the hospital perform a postmortem examination for all the patients included in this study?
I found many bizarre characters in table 1 such as ^, $ and @. These characters are also observed in line 169~175. Please correct them.
In table 3, the unit of laboratory parameters are missing. Please add them. Besides, it would be friendly to readers by listing the reference numbers of each indicator.
Author Response
Dear Reviewer, Thank you for the suggestions. Please see our reply.
- Table 1. According to their data, many patients had more than two comorbidities. It would be necessary to make a statistic on this issue.
Reply: As suggested by the respected reviewer, Table 5 has been added to see the association of up to 2 comorbidities and more than 2 comorbidities with the outcome. Please see page 9 lines 209-211.
- How were the causes of death of patients diagnosed? Did the hospital perform a postmortem examination for all the patients included in this study?
Reply: Cause of death was declared as cardiopulmonary arrest generally for dying patients although postmortem examination was not done for any patient. Please see page 3 lines 133-134
- I found many bizarre characters in table 1 such as ^, $ and @. These characters are also observed in line 169~175. Please correct them.
Reply: As suggested the bizarre characters have been corrected.
- In table 3, the unit of laboratory parameters is missing. Please add them. Besides, it would be friendly to readers by listing the reference numbers of each indicator.
Reply: As suggested the units for lab parameters have been added in table 3.